# Communities of Digger Wasps (Hymenoptera: Spheciformes) along a Tree Cover Gradient in the Cultural Landscape of River Valleys in Poland

**DOI:** 10.3390/insects15020088

**Published:** 2024-01-29

**Authors:** Piotr Olszewski, Tim Sparks, Lucyna Twerd, Bogdan Wiśniowski

**Affiliations:** 1Natural History Museum, Faculty of Biology and Environmental Protection, University of Łódź, Kilińskiego 101, 90-011 Łódź, Poland; 2Department of Zoology, Poznań University of Life Science, Wojska Polskiego 71C, 60-625 Poznań, Poland; thsparks@btopenworld.com; 3Department of Environmental Biology, Kazimierz Wielki University, Ossolińskich 12, 85-093 Bydgoszcz, Poland; l.twerd@ukw.edu.pl; 4Institute of Agricultural Sciences, Land Management and Environmental Protection, University of Rzeszów, Zelwerowicza 8B, 35-601 Rzeszów, Poland; bwisniowski@ur.edu.pl

**Keywords:** Spheciform wasps, insect communities, distribution, food base, phenology

## Abstract

**Simple Summary:**

A study of digger wasps was carried out along a gradient of forest stands in river valleys in northern Poland. One hundred and thirty six species were recorded, representing over half of all Polish digger wasp species. These included 30 species on the Red List of Endangered Animals in Poland. Some additional information on prey species and food plants was also collected. Natural succession to woodland was associated with a change in species composition, but principally with a decline in species richness and diversity.

**Abstract:**

This study of digger wasps (Hymenoptera: Spheciformes) was carried out in the cultural landscape of the Drwęca, Lower Vistula, and Warta river valleys in northern Poland during 2011–2013. The study was undertaken on sites representing a succession gradient from dry grasslands to high levels of tree cover which we hypothesised would influence the structure of digger wasp communities. During our research additional information on flower use, insect prey, and phenology was also recorded and is reported here, revealing dependencies between woodland cover and both the prey and nesting types of digger wasps. A total of 136 species were recorded, i.e., nearly 56% of all Spheciformes species recorded from Poland. Among the species collected, 30 were on the Red List of Threatened Animals in Poland. Most endangered species were recorded in psammophilous grasslands, which are open habitats, and the least in mesic sites. These results significantly update the known distribution of the digger wasp in northern Poland. Knowledge on the biology of digger wasps in Poland is also supplemented by information on the feeding of larvae of 14 species and information on food plants visited by imago digger wasps. The results of our research confirm the correlations between the increase in forest cover and the number of digger wasp species.

## 1. Introduction

Thermophilic grasslands, strongly insolated and rich in nectar plants, are attractive habitats for many species of digger wasps (Hymenoptera: Spheciformes) [1,2,3,4,5,6]. Such grasslands often develop in the valleys of rivers and other watercourses and in moraine areas. Unfortunately, in many European countries, such habitats have declined in terms of both area and in the diversity of their plant cover as a consequence of woodland succession [7]. In Poland, these changes also arise from active afforestation of open arid areas [8].

Often, such areas located in cultural landscapes serve as refugia for many animals. For this reason, this type of landscape is often subject to natural valorisation in order to determine the degree of naturalisation of its individual components. Open, xeric grasslands are not a climax community and without human intervention will go through natural succession towards a woodland habitat. Here, we examine how varying levels of woody cover may affect digger wasps which need abundant nutritional resources and safe nesting sites [1,2]. The attractiveness of these modified habitats for entomophages is still poorly known, and the assessment requires a zoocenological analysis. Information about digger wasps from xeric environments has most often been derived from faunistic-ecological studies [9,10,11,12,13]. The current literature suggests that the nesting biology of most species of digger wasps is poorly known [2,14,15]. Due to the sparsity of knowledge, it is necessary to collect data on the distribution and ecology of digger wasps. To date, research on the ecology of Polish digger wasp communities has been carried out in anthropogenic green areas of the Mazovian Lowlands [16,17]; in the lime–oak–hornbeam and thermophilous oak forests of the Mazovian Lowlands [18]; on predatory Aculeata (Hymenoptera) in moist meadows in the Mazovian Lowlands [19]; in suburban and urban habitats with varying degrees of anthropogenic pressure [16,20]; in the succession stages of the pine forests of Puszcza Białowieska [21]; in various habitats of the Łódź Uplands [22] and the city of Łódź [23]; in Augustów Primeval Forest [24], Ojców National Park [25], and Świętokrzyski National Park [26]; in the forest areas of the Toruń Basin [27]; on post-agricultural areas and fallow fields in the Kampinos National Park [5,10]; in soda ash wasteland which originated as a by-product of industrial processes [28]; and in sand quarries undergoing ecological succession [29].

The present study aims to determine the species composition of digger wasp communities (Spheciformes) in dispersed river valley environments in northern Poland. We tested the hypothesis that natural succession of vegetation resulting in increased woody cover decreases the diversity of digger wasp communities.

## 2. Materials and Methods

### 2.1. Study Area and Sites

The study was carried out in three river valleys (Drwęca, Lower Vistula, and Warta) located in the North European Plain within northern Poland. Research material was collected from nine sites (Figure 1) within this area. The boundaries of the sites were determined from 1:10,000 cartographic maps.

PT: Papowo Toruńskie (53°4′16″ N, 18°41′43″ E): An area of 3.7 ha in the cultural landscape at the edge of the Toruń Basin of the Vistula valley comprising a mosaic of arable fields and pastures, meadows, and small ponds. Woodland vegetation occurrs along roads with thermophilous vegetation, with an admixture of segetal and ruderal plants. The woodland vegetation cover is 0.9% of the site (Figure 1 PT).

SI: Sierakowo (53°10′18″ N, 18°52′20″ E): An area of 0.9 ha of the upland zone at the edge of the Drwęca valley. The site is made up of a sandy loamy esker, covered with turf and additional segetal and ruderal species. The cover of woodland vegetation is 6.7% (Figure 1: SI).

GR: Grudziądz (53°28′20″ N, 18°44′20″ E): An area of 3.3 ha on the edge of the lower Vistula Valley. The slopes of the valley are covered with grassland and ruderal vegetation, and the upper parts with trees and shrubs. The woodland cover is 8.8% (Figure 1: GR).

ZP: ‘Zbocza Płutowskie’ Reserve (53°17′27″ N, 18°22′44″ E): An area of 3.7 ha on the slopes of the lower Vistula valley covered by xerothermic grasslands, shrublands, and oak hornbeam forests with an admixture of *Robinia pseudoacacia* L. The woodland cover is 15.1% (Figure 1: ZP).

GN: ‘Gorzowskie Murawy’ Reserve North (52°43′57″ N, 15°10′45″ E): An area of 2.1 ha of thermophilic sandy grasslands in the Warta valley. The woodland cover is 37.0% (Figure 1: GN).

TP: Toruń-Poligon artillery training ground (52°57′26″ N, 18°34′10″ E): An area of 3.3 ha of the inland dunes of the Toruń Basin of the Vistula valley. The site comprises numerous heaths and grasslands that cover dunes and is diversified by a succession of pine–birch forests, with an admixture of *Populus tremula* L. The woodland cover is 43.2% (Figure 1: TP).

TG: Toruń-Glinki (52°58′8″ N, 18°33′18″ E): An area of 2.8 ha of the inland dunes on the central terraces of the Toruń Basin in the Vistula valley which is dominated by heathlands and sandy grasslands. The woodland cover is 52.6% (Figure 1: TG).

GE: ‘Gorzowskie Murawy’ Reserve East (52°43′34″ N, 15°10′48″ E): An area of 1.2 ha of pure and dry pine forest margin and sandy grasslands in the Warta valley. The woodland cover is 54.0% (Figure 1: GE).

EL: Elgiszewo (53°4′23″ N, 18°55′45″ E). An area of 1.2 ha in the Drwęca river valley within a forested gravel pit. The site is overgrown with sandy grasslands. The woodland cover is 60.3% (Figure 1: EL).

The sites (Figure 1) described above are arranged in order of increasing proportion of woodland cover (bushes and trees). The degree of woodland cover of the site was estimated from the percentage of the area devoid of dense tree stands and bushes. Calculations were made in Corel Draw X4.

Information that was recorded sporadically due to a small number of species was not included in the statistical analysis; these data are identified in parentheses in Appendix A. To characterize nesting preferences, species were divided into two guilds [30]:(i)endogeic species, digging nests in the soil and occupying existing cracks in the soil;(ii)hypergeic species, nesting above the ground in woods or plant shoots.

Kleptoparasitic species were excluded from the division and analysis of nesting preferences, as they do not build their own nests. Additionally, species which both dig nests in the soil and nest above the ground (endogeic–hypergeic) were also treated separately.

Threat categories from the Red List of Threatened Animals in Poland [31] were adopted from Olszewski et al. [32]. The results of this research are supplemented with literature data on nesting methods [30,33] and systematic prey items [2,33] in Appendix A.

### 2.2. Digger Wasp Collection

Wasps were collected at the study sites from the end of April to the end of October, i.e., during their period of active nesting (with no or very little wind (<3 Beaufort scale), less than 30% cloud cover, and temperatures in summer exceeding 16 °C) in 2011–2013. A basic sample was the number of digger wasps caught during 30 min of transect fieldwork using a hand net. The number of samples per site varied slightly because of changing weather conditions and the distance (travel time) between sites. Samples were taken in the most attractive environments on the study sites (often on plants, on open sandy surfaces, etc.). Specimens were mostly collected using an entomological net and then killed with ethyl acetate. For the most abundant and characteristic species (*Bembecinus tridens* and *Ammophila campestris*), no specimens were caught, and their abundance was counted on the transect. Their abundance was estimated only while walking, without stopping from site A to site B on the transect. The chances of double counting one of these individuals were considered low.

To avoid overlap, the sampled areas within sites were a minimum distance of 100 m from each other. Digger wasp species, except as noted above, were identified using a PZO MST-130 stereomicroscope, all by the first author. The sampling was repeated at least three times per month at each site. When adult digger wasps were observed feeding, or capturing and delivering larval food to their nests, this information on food and prey species was also recorded whenever possible. Prey of digger wasps were identified by appropriate specialists (see Acknowledgements). Flight periods of the digger wasps were estimated from records made in all sites and all study years (Appendix A). Plant names are based on Mirek et al. [34]. Identification of digger wasps to species level in the field and of collected material was based on the keys of Dollfuss [35]; Bitsch and Leclercq [36]; Bitsch et al. [37,38,39]; and Jacobs [40]. The material is deposited in the collection of the first author.

The communities were considered a time–space set of populations of species competing for environmental resources, i.e., sources of food (flowering plants, honeydew, and insect prey) and nesting places.

### 2.3. Statistical Analysis

For each site, the total number of specimens, total number of species, and Shannon’s diversity index were calculated.

The nature of the relationship between the prey and nesting types of digger wasps (both expressed as a percentage of all species) and woodland cover was investigated using Pearson correlations. Similarly, the number of specimens, number of species and Shannon index were compared to woody cover using Pearson correlations. Calculations were made using PAST4.0 [41] and Minitab19.

## 3. Results

Information on the species captured at each site, species phenology, flower visits, and prey supplied to larvae is summarised in Appendix A. During the three-year study, 2916 specimens of digger wasps were collected and a total of 136 species were recorded. The following genera were represented by the highest number of recorded species: *Crossocerus* (19 species), *Oxybelus* (8 species), *Ectemnius*, and *Pemphredon* (7 species each).

### 3.1. Dominant Species

The most abundant species of digger wasps were *Bembecinus tridens* (Fabr.) (7.8%), *Diodontus minutus* (Fabr.) (6.6%), *Oxybelus bipunctatus* Oliv. (4.7%), *Mellinus arvensis* (L.) (3.9%), *Cerceris rybyensis* (L.) (3.3%), *Tachysphex obscuripennis* (Schenck) (3.0%), *Oxybelus uniglumis* (Panz.) (2.4%), *Miscophus ater* Lep. (2.2%), and *Alysson spinosus* (Panz.) (2.1%). All of these species build nests underground and prefer sunlit sandy areas [1,2]. The particularly high domination of *B. tridens* resulted from favourable habitat conditions (sandy grasslands) at the SI, GR, GN, TP, TG, and GE sites.

### 3.2. Habitat Preferences

The highest number of endogeic species and their kleptoparasites were found in psammophilous grasslands (Figure 2) in SI (50 endogeic, 24 hypergeic, 1 endogeic/hypergeic and 4 kleptoparasite), GN (40 endogeic, 3 hypergeic, 1 endogeic/hypergeic and 1 kleptoparasite), TP (38 endogeic and 3 hypergeic), and TG (41 endogeic, 6 hypergeic, 1 endogeic/hypergeic and 1 kleptoparasite) (Figure 2, Appendix A). Fewer species were associated with forest habitats, woodland or bushes (Figure 3, Appendix A), characterized by a higher soil moisture. Psammophilous grasslands are open habitats that usually occur in the form of low and loose grass communities with a tussocky structure and a rich and diverse vascular flora, enabling the establishment of underground nests. The largest number of hypergeic species was found in PT (31 hypergeic, 24 endogeic, 3 endogeic/hypergeic, 1 kleptoparasite) and SI (24 hypergeic, 50 endogeic, 1 endogeic/hypergeic and 4 kleptoparasites), ZP (16 hypergeic, 34 endogeic, 1 kleptoparasite), and GR (15 hypergeic, 33 endogeic and 2 kleptoparasite), where the wooded areas with bushes were surrounded by arable fields, meadows, wasteland and farm buildings. Specific habitat characteristics also correspond to the food preferences of the larvae. In xeric sunny habitats (TG and TP), species preying on Orthoptera dominated (Figure 3, Appendix A). In mesic habitats (PT, SI, GR and ZP), species preying on Diptera dominated (Figure 3, Appendix A).

### 3.3. Rare Species

Among the collected digger wasps, 30 species were rare or very rare and threatened in Poland. Digger wasps recorded during research represented the following categories of threat:CR (critically endangered): *Stizus perrisi* Dufour;EN category (Endangered): *Mimumesa littoralis* (Bond.);VU category (vulnerable): *Ectemnius fossorius* (L.) and *Miscophus postumus* Bisch.;NT category (near threatened): *Lestica subterranea* (Fabr.), *Mimumesa beaumonti* Lith, *Harpactus pulchellus* (Costa), *Harpactus leavis* (Latr.), *Cerceris flavilabris* (Fabr.);LC category (Least Concern): *Tachysphex fulvitarsis* (Costa), *Tachysphex psammobius* (Kohl), *Oxybelus variegatus* Wesm., *O. argentatus* Curt., *Nysson niger* Chevr., *Miscophus spurius* (Dahlb.), *Miscophus niger* Dahlb., *M. concolor* Dahlb., *L. alata* (Panz.), *Harpactus elegans* (Lep.), *Gorytes fallax* Handl., *Dryudella pinguis* (Dahlb.), *Crossocerus tarsatus* Shuck., *C. cinxius* Dahlb., *Bembix rostrata* (L.) and *Bembecinus tridens* (Fabr.);DD category (data-deficient): *Astata kashmirensis* Nurse, *Crossocerus styrius* Kohl., *Miscophus ater* Lep., *Tachysphex tarsinus* (Lep.) and *Passaloecus brevilabris* Wolf.

The highest number of species from the Red List was recorded in SI and TP (14 species), whereas the smallest numbers were at PT and ZP (8 species) (Appendix A).

### 3.4. The Influence of Woodland Cover

Increasing woody cover was associated with a greater contribution of endogeic species (r = 0.794, *p* = 0.011) and a reduced contribution of hypergeic species (r = −0.764, *p* = 0.017); no significant relationship was found for the other two groups (Figure 2).

There was an increase in the contribution to prey of three orders/guilds (Lepidoptera, r = 0.962, *p* < 0.001; Orthoptera, r = 0.741, *p* = 0.022; Blattodea, r = 0.727, *p* = 0.027) and a decline in the contribution of two (Diptera, r = −0.904, *p* = 0.001; Hemiptera-Aphidoidea, r = −0.686, *p* = 0.041) with increasing woody cover (Figure 3). There was no significant relationship with the remaining five orders/guilds.

The natural succession of vegetation resulting in increased woody cover was associated with a lower diversity of digger wasp communities, as assessed by both the number of species and by Shannon’s diversity index (Figure 4).

## 4. Discussion

The 136 species recorded represent 56% of the digger wasp species known in Poland [32]. Digger wasps have a wide spectrum of nesting strategies [2]. The success of nesting of certain species, both individually and in nest groups, depends on the availability of suitable habitat conditions. The food spectrum of a community is largely based on the habitat preferences of its individual species [5]. Differences in the total number of species and individuals at specific sites are mainly from habitat differences in abiotic and biotic characteristics. It was observed that the digger wasps preyed on spiders and insects (from at least six orders) at all sites, which, considering the habitat requirements of their potential prey, indicates the high habitat diversity (both abiotic and biotic) of the sites.

During the studied years, digger wasps were mostly observed from early May to early September. The phenology of most of the digger wasps at individual sites differed slightly from the literature data [35]; species were usually observed 2/3 weeks later, especially at the North and East ‘Gorzowskie Murawy’ Reserves (Appendix A).

The food of adult digger wasps was determined by the characteristics of a given area. In psammophilous grasslands, digger wasps were usually observed directly on sand or near plants such as *Scleranthus perennis*, *Sedum* spp., and *Thymus serpyllum*. In mesic habitats, with some trees and shrub vegetation, digger wasps were observed looking for honeydew on leaves of trees, e.g., *Tilia cordata*, *Acer campestre*, and *Ulmus minor*, or hunting prey for their larvae.

The highest number of plants visited by digger wasps was recorded in woodland vegetation and the smallest in xerothermic grasslands [27]. It seems that the development of shrub vegetation is particularly beneficial for the digger wasp fauna because it promotes not only species associated with open areas, but also stenotopic species (Appendix A, Figure 2). However, it should be expected that with spontaneous succession towards forest vegetation, the species diversity of digger wasps will diminish [25,27,28]. Tropek et al. [42] observed a similar trend in industrial waste heap habitats where the number of Aculeata decreased with the simplification of the habitat structure due to succession. As in the industrial example above, thermophilic species that nest below ground and are associated with open habitats predominate at selected sites [5]. The attractiveness of open anthropogenic habitats has also been confirmed for other groups of insects. It was shown that, for example, compared to grasslands, railway embankments were richer in terms of both numbers and species diversity of bees and butterflies [43]. As a result of our research in the valleys of northern Poland, we have shown that inland dunes were especially attractive to digger wasps building nests underground, and they were also attractive to their kleptoparasites.

From the species summary, and taking into account habitat conditions, it is likely that the diversity of species was determined by the heterogeneity of the studied environment, based on, inter alia, vegetation, terrain, soil type and open areas, and we here focus on associations with woody cover (Figure 4). Correlation analysis showed that the high numbers of species preying on Orthoptera and Blattodea are characteristic of thermophilic environments on sandy soils (Appendix A, Figure 3); these environments are characteristic for endogeic species, for example, the genera *Ammophila* Kirby, *Bembecinus* Costa, and *Tachysphex* Kohl. Interpreting a positive correlation in this case depends on numerous biotic and abiotic factors, the most important of which is the presence or absence of larger or smaller sandy areas in the forest environment.

However, the high proportion of species that hunt for aphids or flies indicates the presence mainly of hypergeic species, which prefer wetter habitats with diverse lush vegetation and environmental heterogeneity. This corresponds well to their negative correlation with woodland cover (Figure 2 and Figure 3). The food preferences of larvae and their nesting types depending on woodland cover are still not well understood, so further detailed research is recommended.

These conditions provided access to a diverse food base and diverse nesting sites. The significance of these factors in the quantitative and qualitative structure of wasp communities confirms the findings of Saure [44,45,46] and Szczepko et al. [47].

These river valleys are an attractive habitat for the digger wasps. The same number of digger wasp species were recorded in Kampinos National Park, which is also rich in habitats located on inland dunes and represents various stages of succession of plant communities, from open sandy grounds of anthropogenic origin (abandoned fields) to various forms of woodlands of differentiated ages [5].

The highest number of threatened species were recorded in psammophilous grasslands, which are open habitats, and the lowest at mesic sites (Appendix A). These findings correspond well to those of all threatened species of digger wasps in Poland [32,33].

The data obtained during the present study substantially update our knowledge of the distribution of digger wasps in northern Poland. Our knowledge of the biology of digger wasps in Poland is also supplemented by information on the larval prey of 14 species and information about food plants visited by groups of digger wasps.

The results of our study confirm the findings of other research that the number of digger wasp species decreases with increasing woodland cover (e.g., [25]). This generally refers to species of open habitats [21]. On the other hand, diversity of habitats increases the biodiversity of invertebrates [5]. The heterogeneity of habitats is important as the factor supporting species richness, as was shown in Ojców National Park, where the highest number of aculeate hymenopteran species were recorded in various ecotones [25]. From the point of the view of the protection of biodiversity, the management of sites of significant value should, most of all, preserve the mosaicity of habitats. This includes active protection of open habitats and prevention of the succession of woodlands.

## Figures and Tables

**Figure 1 insects-15-00088-f001:**
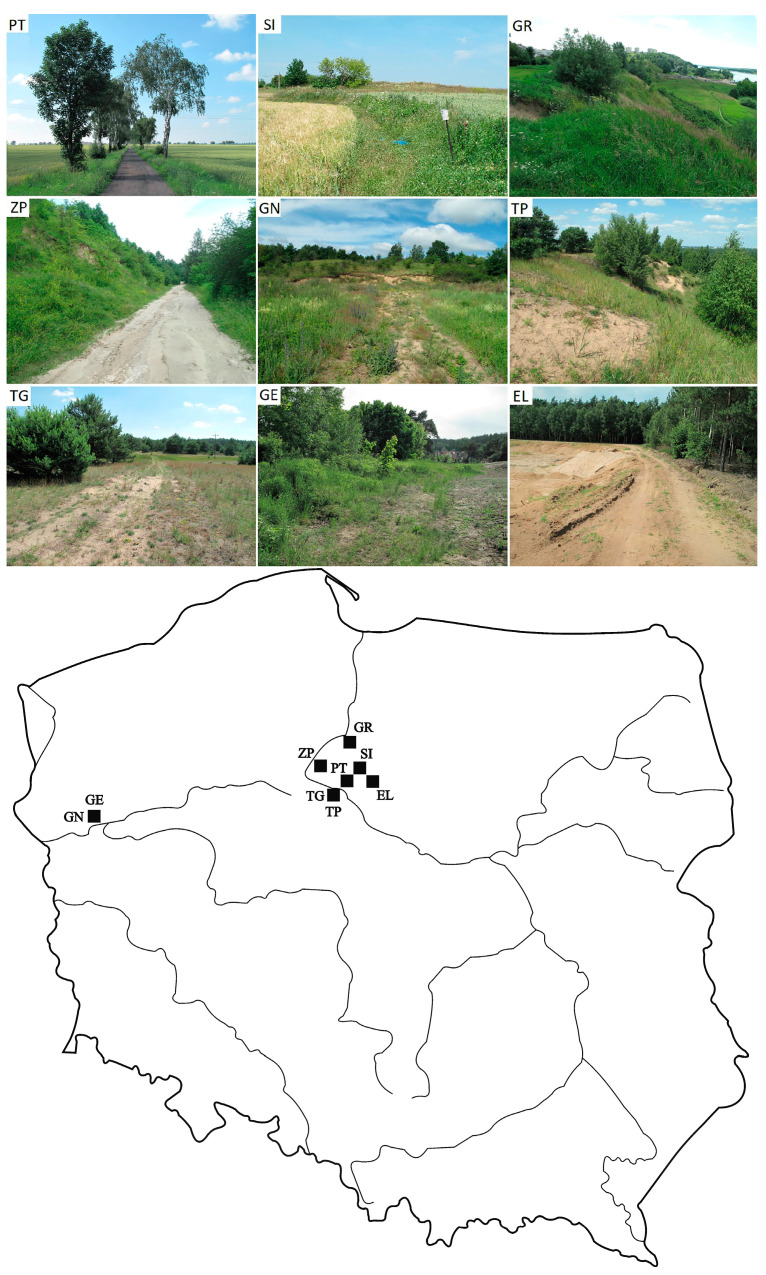
Study sites: PT—Papowo Toruńskie; SI—Sierakowo; GR—Grudziądz; ZP—‘Zbocza Płutowskie’ reserve; GN—‘Gorzowskie Murawy’ reserve North; TP—Toruń-Poligon artillery; TG—Toruń-Glinki; GE—‘Gorzowskie Murawy’ reserve East; EL—Elgiszewo.

**Figure 2 insects-15-00088-f002:**
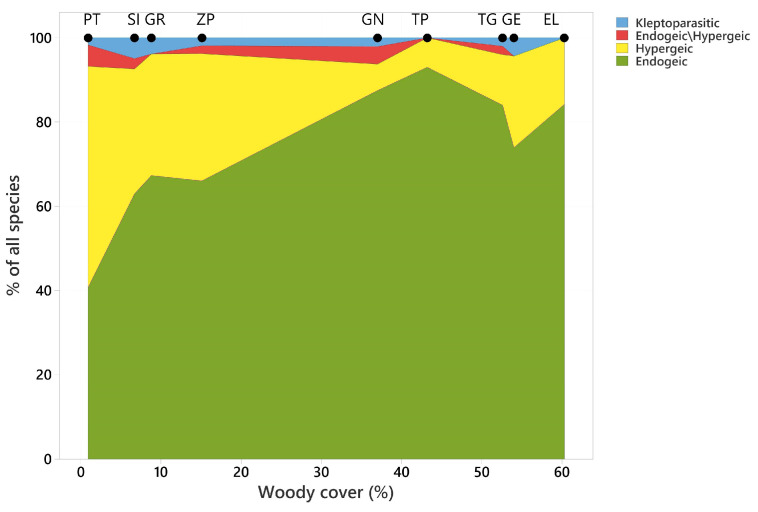
The contribution of four guilds to overall species richness in relation to increasing woody cover. Values of woody cover at survey locations are indicated by solid symbols at the top of the graph.

**Figure 3 insects-15-00088-f003:**
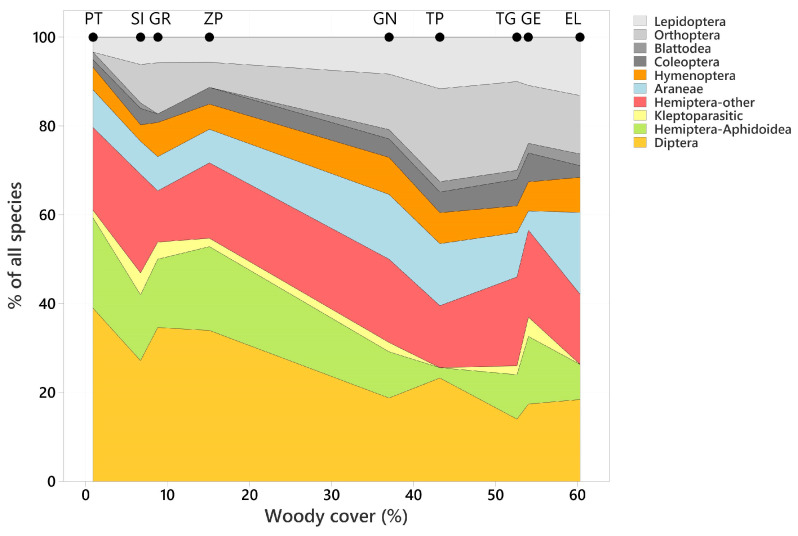
The contribution of ten orders/guilds to overall species richness of prey in relation to increasing woody cover. Values of woody cover at survey locations are indicated by solid symbols at the top of the graph.

**Figure 4 insects-15-00088-f004:**
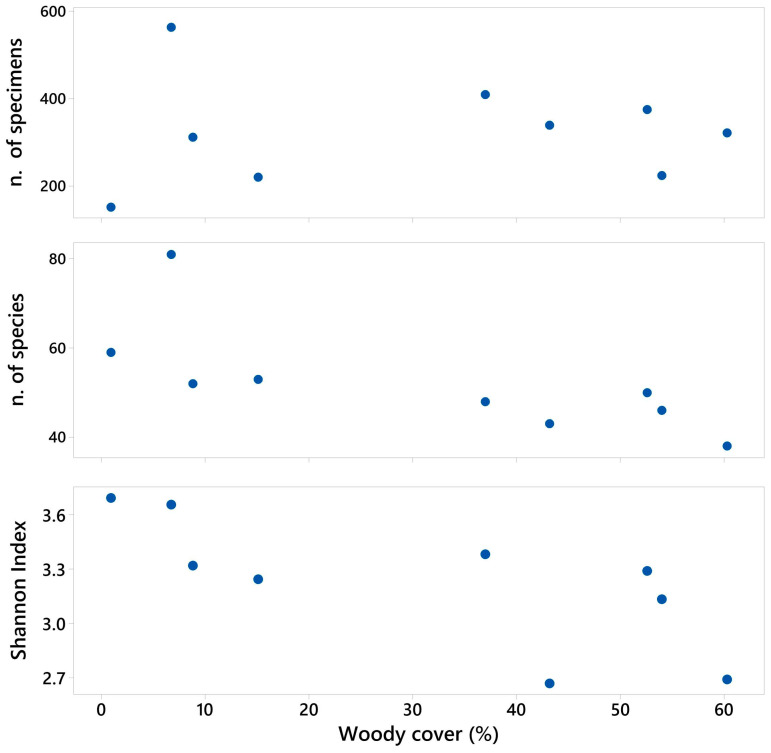
Scatterplots showing the number of individual specimens (r = 0.032, *p* = 0.934), the number of species (r = −0.724, *p* = 0.028), and Shannon’s diversity index (r = −0.734, *p* = 0.024) plotted against woody cover.

## Data Availability

The data that support the findings of this study are available from the corresponding authors upon reasonable request.

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
