# Peer review of "Communities of Digger Wasps (Hymenoptera: Spheciformes) along a Tree Cover Gradient in the Cultural Landscape of River Valleys in Poland"

_insects, 2024, doi:10.3390/insects15020088_

Round 1

Reviewer 1 Report

Comments and Suggestions for Authors

The manuscript "Communities of digger wasps (Hymenoptera: Spheciformes) along a tree cover gradient in the cultural landscape of river valleys in Poland" examines the diversity of digger wasp, the prey they collect, and the regions in northern Poland where they can be found. The nesting and foraging behavior of these wasps is generally understudied, particularly in comparison to social wasps and parasitic wasps. The data collected for this study will be of interest to solitary and social insect researchers (particularly bee and wasp researchers), as well as insect ecologists in general. 

Overall, I found the manuscript interesting. The introduction was a bit brief. Some information on the ecological requirements of digger wasps was provided, but a bit more would be helpful. There were a few aims that were listed in the introduction that were not addressed in the results, and a bit more detail could have been added to some of the methods. I've provided some suggestions below on how this might be addressed. 

Aims: Please clarify whether all of the aims are specific to this one population in northern Poland, or if just the first aim is relevant to that region. Consider adding a bit more to the intro as to why these populations were important to study.

Aim 1: to determine species composition of digger wasp communities. While the % of species and number of species was examined across degrees of woody cover, the digger wasp community composition for each site was only listed in the supplement. Please consider providing an analysis on community composition across sites in the results.

Aim 2: determine the phenology of digger wasps. Does phenology vary across species or populations? Is the population in northern Poland unique or representative of digger wasp populations globally? Given that it was one of the primary aims of the study, the methods used to determine phenology of digger wasps could use more description. The results for this aim seem only to be presented in the supplement table, and it's not clear what the patterns where (if there were any). Please either remove this from the aim or include phenology in the results. The discussion states (line 249-50): digger wasps were mostly observed from early May to early 249 September." Again - these data were not included in the ms, and would have been valuable.  

Aim 3: to gather information about plants visited by adults. These data were not provided in the results (nor the supplemental table). 

Table Supplement: A lot of information is provided "in Table" (which is actually a supplement). Please refer to this in the text as "Supplement" and note that it is just the raw data (it does not show patterns or trends). This was confusing while reading as there was no table in the main manuscript.

Minor comments:
"Digger wasps" are only specified as from the Family Sphecidae. However, in the results (line 170), data from Tachysphex is provided (family: Crabronidae). I didn't check all of the species provided, but please make sure that it is clear what families are being investigated in this study.

Line 53-4: wasps. "To date, research on the ecology of Polish digger wasp communities has been investigated..." I like the review of this literature, but this does not convey a sparsity of knowledge. Do you mean sparsity overall, and you are focusing on Polish locations?

Line 58: "(20,16]" should be "[20,16]"

Line 132: "[3134]" should be "[31,34]"

Line 145: Were the wasps killed before being identified? Were they anesthetized in some way? If ID'd in the field, were they released? If released, did you do anything to ensure you didn't count the same wasp on a second visit?

Line 146 & 155: "lead author" and "first author" are referenced here. If "first author" is meant for both, then just state that in both places. Else, please use last names if it is important to distinguish who among the author list did the work. 

Line 152: "Plant names..." this seems to be in the wrong section. Which plants were identified? How were those data used in the results?

Lines 197-210: This information might be better represented as a table rather than bulleted points.

Line 215: "There was an increase in the contribution to prey of three taxa..." When first reading this, based on the results, I thought 'taxa' should be changed to "Order". However, based on Fig 3, I realized that "Cleptoparasitic" was included, and it was not clear what 'taxa' that represents. Lines 128-9 state: "Kleptoparasitic species were excluded from the division and analysis of nesting preferences, as they do not build their own nests." Please clarify.

Line 221: Figure 4 shows two sets of results, but only one result is described in the text. 

Lines 274-6: "From the species summary, and taking into account habitat conditions, it can be seen that the number of species was determined by the heterogeneity of the studied environment, based on, inter alia, vegetation, terrain, soil type and open areas (Figure 4)." Figure 4 only includes % woody cover. There is nothing about vegetation type presented in the results.

Figure 3: Given that correlative analyses were used for these data, a stacked figure might not be the best way way to illustrate the trends. 

Figure 3: Consider re-ordering the prey taxa alphabetically or taxonomically (Araneae should be separate, as should "Cleptoparasitic")

Figure 4: Consider adding lines to scatterplots with a significant slope. Also, since there were multiple collections at this site, could this be represented as an average +/- SE? Please label top & bottom as A & B and describe in figure caption.

Supplement table still has tracked changes. Please also include more details about what is in the table (What do the codes under "nesting" mean? Please explain why endangered categories are listed under the "species" column, and what those abbreviations mean).

Author Response

Reviewer 1

The manuscript "Communities of digger wasps (Hymenoptera: Spheciformes) along a tree cover gradient in the cultural landscape of river valleys in Poland" examines the diversity of digger wasp, the prey they collect, and the regions in northern Poland where they can be found. The nesting and foraging behavior of these wasps is generally understudied, particularly in comparison to social wasps and parasitic wasps. The data collected for this study will be of interest to solitary and social insect researchers (particularly bee and wasp researchers), as well as insect ecologists in general.

Overall, I found the manuscript interesting. The introduction was a bit brief. Some information on the ecological requirements of digger wasps was provided, but a bit more would be helpful. There were a few aims that were listed in the introduction that were not addressed in the results, and a bit more detail could have been added to some of the methods. I've provided some suggestions below on how this might be addressed.

Aims: Please clarify whether all of the aims are specific to this one population in northern Poland, or if just the first aim is relevant to that region. Consider adding a bit more to the intro as to why these populations were important to study.

Only the first aim is relevant to that region. Text modified.

Aim 1: to determine species composition of digger wasp communities. While the % of species and number of species was examined across degrees of woody cover, the digger wasp community composition for each site was only listed in the supplement. Please consider providing an analysis on community composition across sites in the results.

We have added Shannon’s diversity index. We tried a number of ordination techniques to establish patterns, but these all proved difficult to interpret, partly perhaps due to limited knowledge of the habiatat requirements of individual species.

Aim 2: determine the phenology of digger wasps. Does phenology vary across species or populations? Is the population in northern Poland unique or representative of digger wasp populations globally? Given that it was one of the primary aims of the study, the methods used to determine phenology of digger wasps could use more description. The results for this aim seem only to be presented in the supplement table, and it's not clear what the patterns where (if there were any). Please either remove this from the aim or include phenology in the results. The discussion states (line 249-50): digger wasps were mostly observed from early May to early 249 September." Again - these data were not included in the ms, and would have been valuable. 

Phenology will likely vary across latitudes and change between years. We have used timings of observations across sites and years to produce first estimates of flight periods. These are listed in Supplementary Table 1 for each species, and shown graphically for the more abundant (and hence, better estimated) species in a new figure.

Aim 3: to gather information about plants visited by adults. These data were not provided in the results (nor the supplemental table).

All data regarding the plants visited by digger wasps were included in the table. We don't have any other data.

Table Supplement: A lot of information is provided "in Table" (which is actually a supplement). Please refer to this in the text as "Supplement" and note that it is just the raw data (it does not show patterns or trends). This was confusing while reading as there was no table in the main manuscript.

Corrected.

Minor comments: "Digger wasps" are only specified as from the Family Sphecidae. However, in the results (line 170), data from Tachysphex is provided (family: Crabronidae). I didn't check all of the species provided, but please make sure that it is clear what families are being investigated in this study.

Taxonomic relationships have recently undergone significant changes in this family. Therefore, the frequently used notation Spheciformes was used throughout the work.

Line 53-4: wasps. "To date, research on the ecology of Polish digger wasp communities has been investigated..." I like the review of this literature, but this does not convey a sparsity of knowledge. Do you mean sparsity overall, and you are focusing on Polish locations?

Nesting biology in Poland and on a global scale is poorly known. In this work I list virtually all important ecological works in recent decades.

Line 58: "(20,16]" should be "[20,16]"

Corrected.

Line 132: "[3134]" should be "[31,34]"

Corrected.

Line 145: Were the wasps killed before being identified? Were they anesthetized in some way? If ID'd in the field, were they released? If released, did you do anything to ensure you didn't count the same wasp on a second visit?

In the case of the most abundant species their numbers were estimated whilst walking. We believe the likelihood that a specimen will have been counted twice was small.

Line 146 & 155: "lead author" and "first author" are referenced here. If "first author" is meant for both, then just state that in both places. Else, please use last names if it is important to distinguish who among the author list did the work. 

Corrected.

Line 152: "Plant names..." this seems to be in the wrong section. Which plants were identified? How were those data used in the results?

All plant species on which digger wasps were collected are presented in the table. These data were not detailed in the results because they are incomplete.

Lines 197-210: This information might be better represented as a table rather than bulleted points.

The information is also included in the Supplementary Table. We believe such a table would take up much more space. We leave the decision to the Editorial staff.

Line 215: "There was an increase in the contribution to prey of three taxa..." When first reading this, based on the results, I thought 'taxa' should be changed to "Order". However, based on Fig 3, I realized that "Cleptoparasitic" was included, and it was not clear what 'taxa' that represents. Lines 128-9 state: "Kleptoparasitic species were excluded from the division and analysis of nesting preferences, as they do not build their own nests." Please clarify.

We have changed the wording to order/guild in the hope this will cover the inclusion of cleptoparasites. The earlier reference is to nesting habits of the digger wasps.

Line 221: Figure 4 shows two sets of results, but only one result is described in the text. Lines 274-6: "From the species summary, and taking into account habitat conditions, it can be seen that the number of species was determined by the heterogeneity of the studied environment, based on, inter alia, vegetation, terrain, soil type and open areas (Figure 4)."

This figure has now been changed to include Shannon’s diversity index and all three correlations are reported in the legend to that figure. The text has been modified.

Lines 274-6: "From the species summary, and taking into account habitat conditions, it can be seen that the number of species was determined by the heterogeneity of the studied environment, based on, inter alia, vegetation, terrain, soil type and open areas (Figure 4)." Figure 4 only includes % woody cover. There is nothing about vegetation type presented in the results.

We have modified the text accordingly

Figure 3: Given that correlative analyses were used for these data, a stacked figure might not be the best way way to illustrate the trends. 

With respect, we feel that a stacked figure does portray changes in composition better than, for example, line graphs.

Figure 3: Consider re-ordering the prey taxa alphabetically or taxonomically (Araneae should be separate, as should "Cleptoparasitic")

The order is to help make the trends more obvious, for example declining groups at the base, increasing groups at the top.

Figure 4: Consider adding lines to scatterplots with a significant slope. Also, since there were multiple collections at this site, could this be represented as an average +/- SE? Please label top & bottom as A & B and describe in figure caption.

This figure has been changed as noted earlier. We have nine basic units (sites) for analysis and as such identifying a pattern with woody cover (via correlation) was felt to be more appropiate that attempting to quantify it (via regression).

Supplement table still has tracked changes. Please also include more details about what is in the table (What do the codes under "nesting" mean? Please explain why endangered categories are listed under the "species" column, and what those abbreviations mean).

Apologies, the tracked changes have been removed, the title expanded, and additional abbreviations added at the base of the table.

Reviewer 2 Report

Comments and Suggestions for Authors

Review of MS – INS_2713350 “Communities of digger wasps (Hymenoptera: Spheciformes)  along a tree cover gradient in the cultural landscape of river valleys in Poland”

General

Il MS is primarily focused on the community of digger wasps and their relation with habitat along a tree cover gradient. The results obtained from the investigations are generally interesting as the results can contribute to providing information on the conservation of some habitats and also provide a contribution on a systematic group that is not widely investigated but interesting from an ecological point of view. The topic as discussed, however, is very complex and often some aspects are not clear and this causes inattention to the reader.

Different explanations and indications on set-up of experiments presented are necessary and more in the different paragraphs.

In particular, the information on sampling as presented is lacking and is not clear, as are the data used to investigate the phenology of the different species.

It is unclear how the other taxa were collected and the relationships to the species found. In these circumstances, for the different groups it would be appropriate to specify the type of sampling. Furthermore, specify better whether the trapping method adopted in your study is the most suitable for the group investigated;

Given the availability of quantitative measures that reflect how many different types (such as species) there are in a data set (a community) why no diversity indices have been calculated;

It is advisable to insert a section to better highlight the statistical analyzes adopted following data collection.

The table, in the word file, in the central part (where the sites are listed) could be reduced into two columns (n= number of samples; site=sampled site);

For the most representative species in quantitative terms, flight curves could be highlighted in relation to the months.

Specific comments

Line 117-119. I don't understand the inclusion of this sentence in this context of the MS;

Line 137-139. Here the authors can provide more information about their data collection. Were linear transects used with the net or were areas with very specific surfaces chosen?

Line147-149. The sentence is not clear to me as it is not clear how these observations were made together with the captures. The sentence needs to be better worded and specific information needs to be provided that can help other scholars such as sampling the group.

Line 151. Which table? Furthermore, it is not clear what the phrase means;

Line 168-174. This information can be provided through a table;

Line 213-214. In materials and methods only two groups were specified in relation to nesting preferences. Understandably, it is better to specify M&M purposes that there are 4 groups;

Line 215-219. These results expressed in this way do not highlight what was actually correlated as the data for other taxa were not specified as they were sampled.

Line 223. In addition to the photos it is advisable to include a map;

Line 235. Figure 4 is not explanatory. For example, have you tried to make a regression between the number of species and forest cover? It is clear that the number of species reduces with the coverage of the forest surface and therefore a reduction in biodiversity linked to this group.

Author Response

Reviewer 2

General

OK paragraphs.

In particular, the information on sampling as presented is lacking and is not clear, as are the data used to investigate the phenology of the different species.

It is unclear how the other taxa were collected and the relationships to the species found. In these circumstances, for the different groups it would be appropriate to specify the type of sampling. Furthermore, specify better whether the trapping method adopted in your study is the most suitable for the group investigated;

The text has been changed to clarify these matters. The phenology is a first estimate of flight periods derived from dates of records across sites and years. As such, they are far from perfect and hence we now emphasise these as estimates. Similarly prey and foodplants were recorded when observed.

Given the availability of quantitative measures that reflect how many different types (such as species) there are in a data set (a community) why no diversity indices have been calculated;

We have now calculated, and present, a Shannon index for diversity and show a significant negative association between it and increasing woody cover.

It is advisable to insert a section to better highlight the statistical analyzes adopted following data collection.

Now added

The table, in the word file, in the central part (where the sites are listed) could be reduced into two columns (n= number of samples; site=sampled site);

We are not totally clear what the reviewer intends here, but believe it is just to summarise numbers and sites where the species occurred. If we understand correctly, we do not feel that this would be an improvement on the current tabulation.

For the most representative species in quantitative terms, flight curves could be highlighted in relation to the months.

Given that most species are represented by a small number of records and that the phenology is intended merely as a first estimate of flight period for these species in Poland, we feel these would imply a greater level of accuracy than is justified. As a possible alternative we offer a new table of the flight periods of the 15 most numerous species (those represented by 50+ individuals).

Specific comments

Line 117-119. I don't understand the inclusion of this sentence in this context of the MS;

The text has been modified

Line 137-139. Here the authors can provide more information about their data collection. Were linear transects used with the net or were areas with very specific surfaces chosen?

Some more detail is provided.

Line147-149. The sentence is not clear to me as it is not clear how these observations were made together with the captures. The sentence needs to be better worded and specific information needs to be provided that can help other scholars such as sampling the group.

These additional recordings were made wherever possible and we hope that the word changing makes this description more acceptable.

Line 151. Which table? Furthermore, it is not clear what the phrase means;

We now refer throughout to Supplementary Table 1, and have made some wording changes to hopefully clarify the flight period estimates.

Line 168-174. This information can be provided through a table;

We feel that these four lines of text currently take up less space than a table would.

Line 213-214. In materials and methods only two groups were specified in relation to nesting preferences. Understandably, it is better to specify M&M purposes that there are 4 groups;

Corrected.

Line 215-219. These results expressed in this way do not highlight what was actually correlated as the data for other taxa were not specified as they were sampled.

With apologies, we do not understand what the reviewer intends here

Line 223. In addition to the photos it is advisable to include a map;

We have added a map.

Line 235. Figure 4 is not explanatory. For example, have you tried to make a regression between the number of species and forest cover? It is clear that the number of species reduces with the coverage of the forest surface and therefore a reduction in biodiversity linked to this group.

We have expanded the figure to include Shannon’s diversity index, and report correlation for each subgraph. It is our intention to establish if a relationship exists, rather than to quantify it through regression which we feel may not be fully justified by limited number of sites.

Round 2

Reviewer 1 Report

Comments and Suggestions for Authors

The revisions to the manuscript are appropriate, and I appreciate where the authors disagreed with some suggestions and their explanations. 

Minor recommendations:

1. In Fig 4, the "titles" of the graphs should actually be presented as the y-axis title. 

2. I recommend the authors remove Aims 2 & 3 from the introduction as they are. It's fine to report the additional data on these factors and discuss them in the results, but given that none of the results directly related to those aims are presented in the main manuscript (they are all in the supplement), it's a bit misleading to suggest that these were aims of the study. The first aim is clear and is analysed thoroughly for publication. 

3. One comment that did not seem to be addressed in the text that could still be considered was with regards to the methods and collection. Please include something in the text that explains whether wasps were collected and killed and the methods used for this.

Line 145: Were the wasps killed before being identified? Were they anesthetized in some way? If ID'd in the field, were they released? If released, did you do anything to ensure you didn't count the same wasp on a second visit?

In the case of the most abundant species their numbers were estimated whilst walking. We believe the likelihood that a specimen will have been counted twice was small.

Author Response

Minor recommendations:

  1. In Fig 4, the "titles" of the graphs should actually be presented as the y-axis title. 

The graph has been revised as suggested.

  1. I recommend the authors remove Aims 2 & 3 from the introduction as they are. It's fine to report the additional data on these factors and discuss them in the results, but given that none of the results directly related to those aims are presented in the main manuscript (they are all in the supplement), it's a bit misleading to suggest that these were aims of the study. The first aim is clear and is analysed thoroughly for publication. 

This has been changed as suggested.

  1. One comment that did not seem to be addressed in the text that could still be considered was with regards to the methods and collection. Please include something in the text that explains whether wasps were collected and killed and the methods used for this.

Done.

Line 145: Were the wasps killed before being identified? Were they anesthetized in some way? If ID'd in the field, were they released? If released, did you do anything to ensure you didn't count the same wasp on a second visit?

In the case of the most abundant species their numbers were estimated whilst walking. We believe the likelihood that a specimen will have been counted twice was small.

Reviewer 2 Report

Comments and Suggestions for Authors

The authors made the appropriate changes to the text following the suggestions of the reviewers, improving the MS in the various sections.

Author Response

The authors made the appropriate changes to the text following the suggestions of the reviewers, improving the MS in the various sections.

We thank the reviewer for their approval.